# Detection of Fungal Diseases in Lettuce by VIR-NIR Spectroscopy in Aquaponics

**DOI:** 10.3390/microorganisms11092348

**Published:** 2023-09-20

**Authors:** Ivaylo Sirakov, Katya Velichkova, Toncho Dinev, Desislava Slavcheva-Sirakova, Elica Valkova, Dimitar Yorgov, Petya Veleva, Vasil Atanasov, Stefka Atanassova

**Affiliations:** 1Department of Animal Husbandry—Non-Ruminants and Other Animals, Faculty of Agriculture, Students Campus, Trakia University, 6000 Stara Zagora, Bulgaria; 2Department of Biological Sciences, Faculty of Agriculture, Students Campus, Trakia University, 6000 Stara Zagora, Bulgaria; genova@abv.bg (K.V.); dinev_sz@mail.bg (T.D.); elica_valkova@abv.bg (E.V.); vka@mail.bg (V.A.); 3Department of Botany and Agrometeorology, Faculty of Agronomy, Agricultural University, 12 Mendeleev blvd, 4000 Plovdiv, Bulgaria; s_sirakova@au-plovdiv.bg; 4Department of Agricultural Engineering, Faculty of Agriculture, Students Campus, Trakia University, 6000 Stara Zagora, Bulgaria; dimitar.yorgov@trakia-uni.bg (D.Y.); pveleva@uni-sz.bg (P.V.); atanassova@uni-sz.bg (S.A.)

**Keywords:** *Aspergillus niger*, *Fusarium oxysporum*, *Alternaria alternata*, prevention, pathogen detection, plant health, spectral analysis

## Abstract

One of the main challenges facing the development of aquaponics is disease control, due on one hand to the fact that plants cannot be treated with chemicals because they can lead to mortality in cultured fish. The aim of this study was to apply the visible–near-infrared spectroscopy and vegetation index approach to test aquaponically cultivated lettuce (*Lactuca sativa* L.) infected with different fungal pathogens (*Aspergillus niger*, *Fusarium oxysporum*, and *Alternaria alternata*). The lettuces on the third leaf formation were placed in tanks (with dimensions 1 m/0.50 m/0.35 m) filled up with water from the aquaponics system every second day. In this study, we included reference fungal strains *Aspergillus niger* NBIMCC 3252, *Fusarium oxysporum* NBIMCC 125, and *Alternaria alternata* NBIMCC 109. Diffuse reflectance spectra of the leaves of lettuce were measured directly on the plants using a USB4000 spectrometer in the 450–1100 nm wavelength range. In near-infrared spectral range, the reflectance values of infected leaves are lower than those of the control, which indicates that some changes in cell structures occurred as a result of the fungal infection. All three investigated pathogens had a statistically significant effect on leaf water content and water band index. Vegetative indices such as Chlorophyll Absorption in Reflectance Index (CARI), Modified chlorophyll absorption in reflectance index (MCARI), Plant Senescence Reflectance Index (PSRI), Red Edge Index (REI2), Red Edge Index (REI3), and Water band index (WBI) were found to be effective in distinguishing infected plants from healthy ones, with WBI demonstrating the greatest reliability.

## 1. Introduction

Aquaponics is a system where aquaculture and hydroponics are integrated into a recirculation system [1]. Aquaponics has a number of advantages such as sustainability, reduced resource consumption, and fewer environmental impacts compared to classical forms of aquaculture [2]. One of the main challenges facing the development of this type of technology is disease control, due on one hand to the fact that plants cannot be treated with chemicals because they can lead to mortality in cultured fish and, on the other hand, to the fact that therapeutic agents that can be used to treat diseases in fish can be absorbed and concentrated by the plants [3].

The absence of soil is also one of the advantages of aquaponic systems, as it excludes contamination and the development of so-called soil-borne diseases. Switching from soil to soilless cultivation in an aqueous solution increases the risk of the expansion of other groups of pathogens, the fungal ones, which can easily spread in the recirculating nutrient solution [4]. Typical examples of fungal pathogens that can lead to serious problems in aquaponics are *Alternaria alternata*, *Aspergillus niger*, and *Fusarium oxysporum*.

*A. alternata* is a fungus that produces mycotoxins and causes toxicity of plant parts and fruits. *A. niger* is the most common species of *Aspergillus* and characteristically causes black mold in fruits and vegetables. *F. oxysporum* is responsible for the wilting of the aerial part of a plant [5]. Early detection of fungal pathogens on lettuce plants using VIR-NIR spectroscopy in aquaponics is essential for minimizing crop losses and optimizing resource use, benefiting the economic viability of aquaponic systems. Additionally, this approach advances our scientific understanding of plant –pathogen interactions in controlled environments, enabling more effective disease management strategies.

In recent years, to avoid pesticide residues in food, intensive worldwide research has been conducted to find environmentally friendly and efficient agents that can combat phytopathogenic fungi [6].

The studies related to the control of phytopathogenic microorganisms in these types of innovative systems are very limited [3,4,7,8,9,10,11]. Most of these studies emphasize the possibility that the suppressive effect of these types of systems is due to naturally occurring beneficial microorganisms in the system [3,4,7,10]. One study investigated different methods (preventive measures—the use of certified disease-resistant plant varieties and cultural practices that keep the plants in optimum conditions to withstand pests and diseases—and ecological approach—maintenance of natural enemies and/or microbial competitors) of treating pathogens in this type of system [8].

An important factor in the combat of fungal pathogens in this system is the prompt detection of their appearance, which could prevent their spread. Prompt detection of fungal pathogens (latent infection) is a crucial factor in combating their presence in this system, as it effectively prevents their spread and subsequent damage. Crop conditions, including diseases, could be estimated based on plants’ spectral characteristics in the visible and near-infrared regions [12,13,14]. Spectral features of plants in the visible region from 400 to 700 nm are connected mainly with the content of pigments. The structure of leaves and their water content define the spectra in the near-infrared region from 700 to 1100 nm. Spectral vegetation indices are developed based on plants’ reflectance at two or more wavelengths designed to highlight a particular property of vegetation. They are used to analyze and to detect changes in plant physiology and chemistry [12,15,16].

The aim of this study was to apply the visible–near-infrared spectroscopy and vegetation indices approach to test aquaponically cultivated lettuce (*Lactuca sativa* L.) infected with different fungal pathogens (*Aspergillus niger*, *Fusarium oxysporum*, and *Alternaria alternata*).

## 2. Materials and Methods

### 2.1. Experimental Design and Characterization of Aquaponic System and Vessels

The effect on vegetative indices obtained through NIR spectroscopy was tested during infection with three fungal pathogen species (*A. alternata*, *A. niger*, and *F. oxysporum*).

The lettuces on the third leaf formation were placed in tanks (with dimensions 1 m/0.50 m/0.35 m) filled up with water from the aquaponics system every second day. The aquaponics system was positioned in the aquaculture base at the Faculty of Agriculture at Trakia University (Stara Zagora, Bulgaria). The nutrient solution was taken from the plant section of the system itself, which had been in operation for 4 months at that point.

The tanks were stored at the Botanical laboratory of the Faculty of Agriculture, where appropriate climatic and light conditions were ensured. Excess water before each refill with a fresh nutrient solution from a 6-month-operating aquaponics system was removed by siphoning. A minimum distance of 20 cm between plants was maintained. In order to prevent anaerobic zones around the roots of the plant, the containers were oxygenated with an aerator.

### 2.2. Microorganisms Studied

This study included reference fungal strains *Aspergillus niger* NBIMCC 3252, *Fusarium oxysporum* NBIMCC 125, and *Alternaria alternata* NBIMCC 109, all of which were obtained from the National Bank for Industrial Microorganisms and Cell Cultures (NBIMCC) in Bulgaria. The fungal strains were stored at 0–4 °C. Prior to use, they were grown on potato dextrose agar (Biolife, Monza, Italy).

The fungal cultures were grown on potato dextrose agar (PDA). Every Petri dish was filled with 20 mL of PDA. After solidification, 0.1 mL of inoculum of the fungal strains (1–2 × 10^4^ CFU/mL) was introduced onto the agar plate surface. An incubation period of 5–7 days at 26–28 °C was maintained. Afterwards, 3–5 g of the infected PDA (hyphal disk) was introduced on the surface of every lettuce plant (*L. sativa*). Ten days later, initial infection with all three pathogens (*A. niger*, *F. oxysporum*, and *A. alternata*) was examined in the plants (Figure 1).

Before infection, another group was established to serve as a control, consisting of uninfected plants (control group). All experimental groups included 10 plants each.

### 2.3. NIR Spectroscopy Measurement

The spectral measurements of the lettuce plants were performed as follows (Figure 1).

The plants subject to spectral study were removed from the aquaponic tanks together with the pots in which they were rooted and placed in other tubs. In this condition, the plants were transferred to the spectral measurement laboratory. Every single plant was taken out from the tanks, measured during about 10–15 min, and put back in it.

Diffuse reflectance spectra of the leaves of lettuce were measured directly on the plants. The measurements were performed with a USB4000 spectrometer (Ocean Optics, Inc., Orlando, FL, USA) in the 450–1100 nm wavelength range. A fiber optics probe for diffuse reflection measurement was used. The probe consists of several optic fibers through which the illuminating light from a halogen lamp was fed to the area of interest and one optic fiber through which the light reflected from the measured surface was delivered to the spectrometer. The recorded spectra represent a dependence of the reflectivity of an area of about 6.5 mm in diameter on the wavelength. Every single spectrum corresponds to a different location on the lettuce leaves. As the plants were not large, all leaves large enough to reliably measure a circular area of about 6.5 mm in diameter were used for measurements. Further, 24 vegetation indices (shown in Table 1) were calculated from every spectrum and their values were analyzed.

### 2.4. Statistical Data Analysis

Statistical data analysis includes multivariate ANOVA, applied to calculate the significant differences between the plants infected with different fungal strains, based on the vegetative indices, most sensitive to the presence of investigated diseases. Significant differences were assumed based on the post hoc multiple comparisons with the Dunnett T3 or Tukey’s test (depending on Levene’s test of equality of error variances) at *p*-value ≤ 0.05. The IBM SPSS Statistics 26.0 (IBM, Chicago, IL, USA) package was used to process the data.

## 3. Results and Discussion

Healthy plants exhibited lower visible reflectance around 470 and 640–670 nm due to the chlorophyll absorption bands and higher near-infrared reflectance. In contrast to the healthy plants, infected plants showed higher reflectance in these areas because of their decreasing chlorophyll content. Changes were also observed in the green area of visible spectra. The reflection from plants infected with *A. niger* and *A. alternata* was lower than that of control plants, showing a change in the color of leaves. In the near-infrared spectral range, the reflectance values of infected leaves were lower than those of the control, which indicates that some changes in cell structures appeared as a result of the fungal infection (Figure 2).

The results of the multivariate ANOVA (Table 2) show that the vegetative indices CARI, MCARI, PSRI, REI2, REI3, and WBI could be used to distinguish plants infected by the pathogen *A. alternata* from the control variant. *A. alternata* is an important phytopathogen that is occasionally found in lettuce crops [33]. Only the vegetative indices PSRI and WBI could be used for distinguishing plants infected by the pathogen *A. niger* from the control variant. It was found that *A. niger* cause over 30% of fungal deterioration in lettuce from Yankaba and Sharada vegetable markets (situated in Nigeria), which made *A. niger* the most prevalent fungal pathogen in lettuce according to the samples studied [34]. The largest numbers of vegetative indices have the ability to distinguish plants infected by the pathogen *F. oxysporum* from the control variant as follows: CARI, Cl_red edge_, Greenness index, MCARI, mNDVI, PI, REI 1, REI 2, REI 3, and WBI. On the basis of an analysis of variance, WBI was found to be the most suitable for finding reliable differences between plants infected with the studied pathogens and those of the control variant. The coefficient of determination of this index is R^2^ = 0.553 (Table 1), i.e., about 55.3% of the variations in the vegetation indices are due to the influence of the pathogen type. According to Mukhtar et al. [35], *F. oxysporum* causes moderate rot on 46.15% of the healthy lettuce samples inoculated with this fungal phytopathogen. This was the highest severity of rot, compared to Rhizopus spp., Rhizoctonia spp., *A. flavus*, *A. fumigatus*, *A. niger*, *F. oxysporum*, and *Mucor* sp. rot in sweet orange, cucumber, and lettuce. This is probably due at least in part to the structure of lettuce, since in cucumber, *F. oxysporum* caused mild rot in 20.63% of the healthy lettuce samples inoculated with this pathogen. These authors found that F. oxysporum caused rot in cucumber more frequently than *A. niger*—20.63% vs. 16.55%, respectively.

As a result of various diseases, including fungal ones, changes in the fluorescence of chlorophyll and, as a result, in the process of photosynthesis have been found [36]. The interaction of fungal pathogens and plants (method of obtaining nutrition, growth type, pigmentation, developmental stage, and disease severity) influenced the changes in spectral characteristics of infected plants. Changes in plant physiology during a pathogen attack can be linked to specific changes in plant reflectance patterns. The plants stop photosynthesizing and performing other assimilatory metabolism processes, and increase respiration and other processes required for defense from fungal pathogens. Although the suppression of photosynthesis and the induction of sink metabolism appear to be common responses to pathogen infections, the effect on sugar levels varies greatly between different plant–pathogen interactions [37]. For example, spectral changes in the region from 500 to 680 nm are observed if a pathogen has influenced the photosynthesis pigments. Pathogens changing the cellular structure of plants have a high influence in the near-infrared region from 700 to 1000 nm.

All three investigated pathogens had a statistically significant effect on leaf water content and water band index. Leaf symptoms of *A. alternata* include round, brown spots with concentric rings. This explains the statistical significance of the differences between the CARI, MCARI, Clred edge, RE2, and RE3 indices, related to chlorophyll content. Zarger et al. [38] found that *A. alternata* infection decreases the content of chlorophyll, carotenoids, ascorbic acid, proteins, and sugar in mango leaves. The same pathogen was found to decrease the lycopene, β-carotene, and total carotenoids but to increase proline content in tomato fruits [39]. The smallest changes in spectral characteristics were observed in plants infected with *A. niger*. Apart from the water band index, we only have statistically significant differences for PSRI. Studies of plant primary metabolism and plant–pathogen interactions based on quantitative imaging of chlorophyll fluorescence showed that photosynthesis was reduced both in cells directly below the fungal colonies and in adjacent cells [40].

*F. oxysporum* colonizes and metastasizes in xylem vessels, and causes systemic yellowing, wilting, and death of infected plants. These changes lead to changes in the spectral characteristics of infected plants both in the visible range and in the range above 700 nm. This explains the higher number of vegetative indices (CARI, MCARI, PI, G, PSRI, mNDVI, RE1, RE2, and RE3) that are statistically significantly different between infected and healthy plants.

Some of the indices allow for distinguishing the type of pathogen.

Expanding upon this point, the presence of the three pathogens—*A. niger*, *A. alternata*, and *F. oxysporum*—brings about different alterations in the spectral characteristics of infected lettuce plants, reflected in the various vegetative indices [41]. These differences suggest that different pathogens affect the physiological and biochemical processes of plants to different extents, leading to alterations in the reflectance spectra [42].

While *A. alternata* affects chlorophyll content and, consequently, coloration and reflectance in the visible range [28], *F. oxysporum* metastasizes in xylem vessels, affecting a broader range of physiological processes, including water content [43]. In contrast, *A. niger* appears to cause the least change in spectral characteristics, primarily influencing the water band index and photosynthesis, reflected in the PSRI [44].

The differences observed in the impact of each pathogen might be attributed to their distinct modes of infection and the corresponding plant responses [45]. For instance, *A. niger*, a common pathogen of many crops, may provoke a more localized response, primarily affecting photosynthesis and resulting in limited changes to spectral characteristics [40].

On the other hand, *A. alternata*, well-known for causing leaf spot diseases, can influence a broader range of physiological processes, as evidenced by the reduction in chlorophyll, carotenoids, ascorbic acid, proteins, and sugar in infected leaves [46]. This drastic change in physiological attributes could significantly alter the reflectance patterns, making it easier to detect this pathogen compared to others [44].

Similarly, *F. oxysporum*, a systemic pathogen known to colonize xylem vessels, can alter plant physiology extensively, leading to systemic yellowing, wilting, and death [35]. These physiological changes would be expected to dramatically alter the plant’s reflectance patterns, making it potentially easier to identify infection with this pathogen using spectral analysis [43].

Interestingly, the distinct spectral changes induced by each pathogen might potentially provide a way to not only detect infection but also identify the type of pathogen involved [40]. Indices like PSRI and WBI, which were significantly different for plants infected by *A. niger* and *A. alternata*, could serve as key indicators for these specific pathogens [30].

Meanwhile, several indices, including CARI, MCARI, PI, G, PSRI, mNDVI, RE1, RE2, and RE3, which were significantly different in plants infected with *F. oxysporum*, could help in the identification of this pathogen [41]. These findings highlight the potential of using spectral analysis and vegetative indices as non-invasive, real-time tools for disease detection and identification in crops [42].

Nonetheless, it is critical to note that these findings are based on controlled experimental conditions, and further research is needed to validate these indices under field conditions where multiple stress factors can influence the spectral characteristics of plants [43]. Also, disease progression stages may present varying spectral signatures and need to be considered in further studies [47].

Moreover, there are specific environmental and genetic factors that could influence these spectral changes and should be taken into account [48]. Changes in light conditions, soil properties, plant nutrient status, and even plant genetic variations could potentially affect plant reflectance patterns and thus the performance of these vegetative indices in disease detection [49].

While this study provides promising results in detecting and identifying specific pathogens based on their distinct impacts on plant spectral characteristics, it also raises several questions. Future research should focus on understanding how these indices behave under different environmental and disease conditions and whether it is possible to develop universal or crop-specific indices for disease detection [48].

Additionally, more advanced techniques, such as machine learning algorithms, could be explored to analyze spectral data and extract more reliable and precise disease indicators [50]. Such methodologies could potentially improve the sensitivity and specificity of disease detection using remote sensing techniques [49].

Finally, the practical application of these findings in real-time disease detection in fields should be explored. This would involve developing cost-effective, portable spectral sensors and integrating them with crop management systems to facilitate timely disease detection and treatment [18].

Our understanding of how different pathogens impact plant spectral characteristics has increased substantially. It has allowed us to leverage this knowledge in developing more precise disease detection and identification techniques. The future of crop disease management will likely heavily rely on the advancements in remote sensing technologies and machine learning algorithms, improving both the speed and accuracy of disease detection [48]. By continuing research in this area, we can potentially revolutionize disease management in agriculture, reducing crop losses and improving global food security.

## 4. Conclusions

In conclusion, this study of reflectance spectra in lettuce plants infected with different pathogens has provided valuable insights into the effects of fungal infections on plant physiology and reflectance patterns. Vegetative indices such as CARI, MCARI, PSRI, REI2, REI3, and WBI were found to be effective in distinguishing infected plants from healthy ones, with WBI demonstrating the greatest reliability. The distinct influences of *Aspergillus niger*, *Alternaria alternata*, and *Fusarium oxysporum* on the reflectance spectra emphasize the potential for spectral analysis in pathogen detection and crop health management. Furthermore, understanding the impact of these fungal pathogens on chlorophyll fluorescence and photosynthesis provides a crucial link to monitoring plant health and productivity. This study underscores the importance of further exploring the potential of remote sensing and spectral analysis in precision agriculture for optimal disease management and productivity.

## Figures and Tables

**Figure 1 microorganisms-11-02348-f001:**
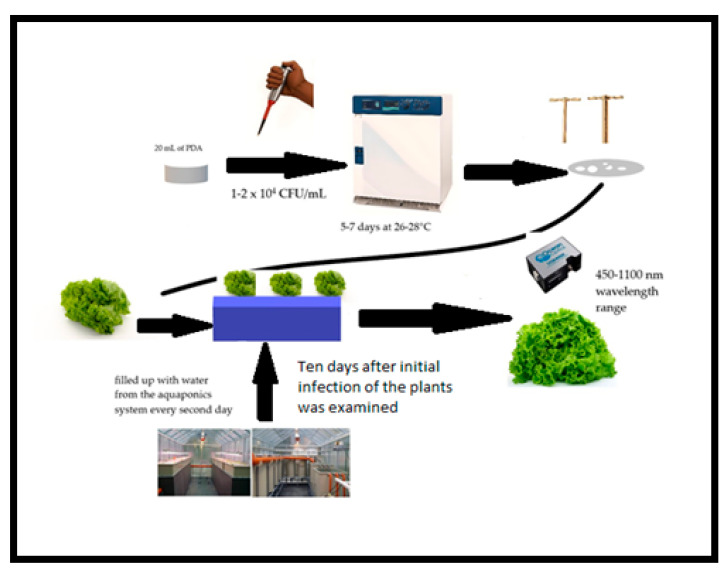
Scheme of experimental design.

**Figure 2 microorganisms-11-02348-f002:**
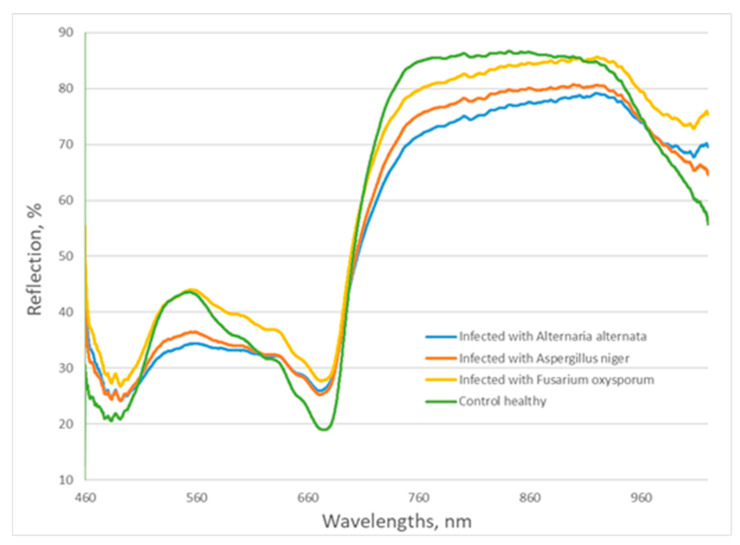
Reflectance spectra of control and infected lettuce plants.

**Table 1 microorganisms-11-02348-t001:** Vegetation indices—definitions.

Index	Definition	References
CARI Chlorophyll Absorption in Reflectance Index	$CARI=\left(R_{700}-R_{670}\right)-0.2\left(R_{700}-R_{550}\right)$	Haboudane et al. [17]
Cl _green_ Clorophyll Index	$Cl_{green}=\frac{R_{760}}{R_{550}}-1$	Gitelson et al. [18]
Cl _red edge_ Clorophyll Index at red edge	$Cl_{red\;edge}=\frac{R_{760}}{R_{714}}-1$	Gitelson et al. [18]
CLSI Cercospora leaf spot index	$CLSI=\frac{\left(R_{698}-R_{570}\right)}{\left(R_{698}+R_{570}\right)}-R_{734}$	Mahlein et al. [19]
CRI Carotenoids Reflectance index	$CRI=\frac{1}{R_{510}}-\frac{1}{R_{550}}$	Gitelson et al. [20]
f_D_ Index of disease	$f_{D}=\frac{R_{500}}{R_{500}+R_{570}}$	Moshou et al. [21]
G Greennes index	$G=\frac{R_{554}}{R_{677}}$	Smith et al. [22]
HI Healthy index	$HI=\frac{R_{534}-R_{698}}{R_{534}+R_{698}}-0.5R_{704}$	Mahlein et al. [19]
MCARI Modified chlorophyll absorption in reflectance index	$MCARI=\left[\left(R_{700}-R_{670}\right)-0.2\left(R_{700}-R_{550}\right)\right]\frac{R_{700}}{R_{670}}$	Daughtry et al. [23]
mNDVI Modified normalized difference vegetation index	$mNDVI=\frac{R_{750}-R_{705}}{R_{750}+R_{705}}$	Tucker et al. [24]
NDVI (1) Normalized difference vegetation index	$NDVI=\frac{R_{845}-R_{665}}{R_{845}+R_{665}}$	Tucker et al. [24]
NDVI (2) Normalized difference vegetation index	$NDVI=\frac{R_{800}-R_{680}}{R_{800}+R_{680}}$	Main et al. [25]
NDVI (3) Normalized difference vegetation index	$NDVI=\frac{R_{800}-R_{670}}{R_{800}+R_{670}}$	Brantley et al. [26]
PI Pigment index	$PI=\frac{R_{677}}{R_{554}}$	Datt [27]
PMI Powdery mildew index	$PMI=\frac{R_{520}-R_{584}}{R_{520}+R_{584}}-R_{724}$	Mahlein et al. [19]
PRI Photochemical reflectance index	$PRI=\frac{R_{531}-R_{570}}{R_{531}+R_{570}}$	Sims and Gamon [15]
PSRI Plant Senescence Reflectance Index	$PSRI=\frac{R_{680}-R_{500}}{R_{750}}$	Merzlyak et al. [28]
REI 1 Red Edge Index	$REI1=\frac{R_{740}}{R_{720}}$	Vogelmann et al. [29]
REI 2 Red Edge Index	$REI2=\frac{R_{734}-R_{747}}{R_{715}-R_{720}}$	Vogelmann et al. [29]
REI 3 Red Edge Index	$REI3=\frac{R_{734}-R_{747}}{R_{715}-R_{726}}$	Vogelmann et al. [29]
SBRI Sugar beet rust index	$SBRI=\frac{R_{520}-R_{513}}{R_{570}+R_{513}}-0.5R_{704}$	Mahlein et al. [19]
SR Simple ratio	$SR=\frac{R_{760}}{R_{695}}$	Gitelson and Merzlyak [30]
TVI triangular vegetation index	$TVI=0.5\left[120\left(R_{750}-R_{550}\right)-200\left(R_{670}-R_{550}\right)\right]$	Broge and Leblanc [31]
WBI Water band index	$WBI=\frac{R_{900}}{R_{970}}$	Wang and Qu [32]

**Table 2 microorganisms-11-02348-t002:** Multivariate ANOVA of the observed plant infections based on the varied types of vegetation indices.

x ± SD					
Vegetation Indices	Classes				R^2^
	Infected with *Alternaria alternata* (n = 8)	Infected with *Aspergillus niger* (n = 9)	Infected with *Fusarium oxysporum* (n = 7)	Uninfected (n = 5)	
CARI	1.975 ± 0.676 ^a^	2.356 ± 0.354	1.854 ± 0.251 ^c^	2.744 ± 0.535 ^a, c^	0.339
Cl_green_	5.146 ± 2.228	3.909 ± 1.051	5.195 ± 1.260	3.457 ± 0.965	0.212
Cl_red edge_	0.817 ± 0.208	0.682 ± 0.120	0.838 ± 0.010 ^c^	0.599 ± 0.133 ^c^	0.309
CLSI	−10.957 ± 0.932	−11.204 ± 0.393	−11.408 ± 0.576	−11.466 ± 0.419	0.099
CRI	−1.661 ± 3.670	−2.825 ± 0.550	−4.117 ± 3.463	−2.604 ± 1.110	0.116
f_D_	8.038 ± 30.228	−2.741 ± 6.650	0.184 ± 2.914	−2.362 ± 6.510	0.077
G	2.800 ± 0.706	3.291 ± 0.581	2.483 ± 0.783 ^c^	3.732 ± 0.580 ^c^	0.333
HI	−2.168 ± 0.436	−2.344 ± 0.174	−2.184 ± 0.193	−2.576 ± 0.307	0.225
MCARI	7.744 ± 3.630 ^a^	10.253 ± 2.614	6.681 ± 3.039 ^c^	13.665 ± 4.427 ^a, c^	0.378
mNDVI	0.486 ± 0.075	0.446 ± 0.040	0.497 ± 0.032 ^c^	0.412 ± 0.056 ^c^	0.285
NDVI (1)	0.921 ± 0.053	0.928 ± 0.017	0.913 ± 0.049	0.930 ± 0.025	0.032
NDVI (2)	0.844 ± 0.043	0.849 ± 0.012	0.837 ± 0.039	0.852 ± 0.018	0.032
NDVI (3)	0.874 ± 0.046	0.880 ± 0.014	0.867 ± 0.042	0.884 ± 0.021	0.038
PI	0.380 ± 0.103	0.315 ± 0.074	0.434 ± 0.119 ^c^	0.273 ± 0.041 ^c^	0.316
PMI	−11.719 ± 4.662	−10.247 ± 0.336	−10.369 ± 0.292	−10.669 ± 0.390	0.066
PRI	−0.825 ± 0.877	−0.436 ± 0.158	−0.619 ± 0.276	−0.405 ± 0.144	0.119
PSRI	0.207 ± 0.014 ^a^	0.215 ± 0.009 ^b^	0.220 ± 0.013	0.232 ± 0.009 ^a, b^	0.397
REI 1	1.382 ± 0.090	1.324 ± 0.052	1.393 ± 0.042 ^c^	1.287 ± 0.059 ^c^	0.314
REI 2	0.678 ± 0.090 ^a^	0.612 ± 0.063	0.682 ± 0.053 ^c^	0.557 ± 0.063 ^a, c^	0.352
REI 3	0.336 ± 0.038 ^a^	0.308 ± 0.026	0.338 ± 0.020 ^c^	0.282 ± 0.029 ^a, c^	0.377
SBRI	1.383 ± 9.315	5.111 ± 16.361	−4.683 ± 6.522	0.840 ± 3.952	0.110
SR	7.739 ± 2.622	6.532 ± 0.912	7.506 ± 1.413	5.999 ± 1.159	0.154
TVI	738.660 ± 46.810	762.604 ± 23.518	757.503 ± 19.201	785.200 ± 25.646	0.218
WBI	1.882 ± 0.063 ^a^	1.929 ± 0.044 ^b^	1.920 ± 0.058 ^c^	2.049 ± 0.046 ^a, b, c^	0.553

Same superscripts within the same row represent significant differences at the level of significance *p* < 0.05, as follows: ^a^—between the group infected with *Alternaria alternata* and all other groups; ^b^—between the group infected with *Aspergillus niger* and all other groups; ^c^—between the group infected with *Fusarium oxysporum* and all other groups; no statistically significant differences existed if there are no superscript letters in the row. R^2^—coefficients of determination based on observed means.

## Data Availability

All data are available from the corresponding author upon request.

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
