# Peer review of "Detection of Fungal Diseases in Lettuce by VIR-NIR Spectroscopy in Aquaponics"

_microorganisms, 2023, doi:10.3390/microorganisms11092348_

Round 1
Reviewer 1 Report
Dear Authors,
Your manuscript is very interesting and relevant for the scientific community. Please edit the manuscript technically according to the instructions for authors.
Best Regards
Author Response
Dear Reviewer,
I wanted to take a moment to express my sincere gratitude for the time and effort you invested in reviewing my manuscript titled "Fungal Diseases Of Cultivated Plants In Aquaponic Systems – Prevention By VIR-NIR Spectroscopy Monitoring." Your insightful comments and constructive criticisms were invaluable in enhancing the quality of our work.
Sincerely,
Sirakov et al.,
Reviewer 2 Report
Fungal Diseases of Cultivated Plants in Aquaponic Systems – Prevention by VIR-NIR Spectroscopy Monitoring by Sirakov et al. seeks to optimize spectroscopy-based disease monitoring in lettuce grown in aquaponic system. This technique can potentially be used to detect plant disease at the very early stage of infection, thus facilitating the adoption of proactive measures for managing diseases. While this work generated variable reflectance spectra that can be used to discriminate infected plants from healthy ones, I firmly believe the data need much more refinement following well-described methods before it can be published. It will be very difficult to replicate this study even by the investigators who work in the same specialty area. It is often very difficult to understand what authors are trying to present starting from the title of the MS. It may be more appropriate to take a title like “Detection of fungal diseases in lettuce by VIR-NIR Spectroscopy”. Materials and Methods should be clearly described so readers understand how the work was done. I suggest adding some color photos to clarify where the lettuce leaves were placed and how USB4000 spectrometer recorded the reflectance spectra. Please follow the journal's editorial style and ensure you use standard English for scientific communications. I highlighted many of the texts in PDF that need to be revised with some suggestions. But the whole manuscript will need attention from someone who is well-versed in English. I want you to revise the inoculation method and show the photos of symptoms if any were due to inoculation. If the symptoms were not visible, that would make this study even more interesting. I also find careless mistakes (highlighted, suggested corrections) all over the MS that must be corrected.

please see above
Author Response
Dear Reviewer,
We wanted to take a moment to express our sincere gratitude for the time and effort you invested in reviewing our manuscript titled "Fungal Diseases Of Cultivated Plants In Aquaponic Systems – Prevention By VIR-NIR Spectroscopy Monitoring." Your insightful comments and constructive criticisms were invaluable in enhancing the quality of our work.
Answers to the questions and comments posed:
This technique can potentially be used to detect plant disease at the very early stage of infection, thus facilitating the adoption of proactive measures for managing diseases. While this work generated variable reflectance spectra that can be used to discriminate infected plants from healthy ones, I firmly believe the data need much more refinement following well-described methods before it can be published. It will be very difficult to replicate this study even by the investigators who work in the same specialty area. It is often very difficult to understand what authors are trying to present starting from the title of the MS. It may be more appropriate to take a title like “Detection of fungal diseases in lettuce by VIR-NIR Spectroscopy”.
The title was corrected based on the reviewer's notes.
Materials and Methods should be clearly described so readers understand how the work was done. I suggest adding some color photos to clarify where the lettuce leaves were placed and how USB4000 spectrometer recorded the reflectance spectra.
Please follow the journal's editorial style and ensure you use standard English for scientific communications. I highlighted many of the texts in PDF that need to be revised with some suggestions. But the whole manuscript will need attention from someone who is well-versed in English. I want you to revise the inoculation method and show the photos of symptoms if any were due to inoculation. If the symptoms were not visible, that would make this study even more interesting. I also find careless mistakes (highlighted, suggested corrections) all over the MS that must be corrected.
The figure clarifying the methodology of the current experiment has been added to the paper, facilitating the study's replication. The manuscript has been revised to align with the journal's editorial style, and the quality of the English in the paper has been significantly improved. All corrections suggested by the reviewer have been addressed. The photos will be added to Supplementary Materials.
Having thoroughly addressed the questions and comments you posed, we sincerely hope that the revised article now meets your expectations. We endeavored to ensure that all your concerns were dealt with comprehensively, and the changes were incorporated in light of your valuable suggestions.
Once again, thank you for your meticulous review and for providing such constructive feedback. We look forward to any further comments or recommendations you might have, and we remain hopeful that the revised submission will meet your approval.
Sincerely,
Sirakov et al.,

Reviewer 3 Report
A scheme of the system is necessary to understand. From the information it is not possible to evaluate if the system was an open or close one.
It is described that plants with three leaves were placed in tanks. In other section it is said that plants in pots were taken out to the system for spectrophotometry measurements. It is confusing. How was the system. A figure may help
The aquatic organisms are not described. How much biomass of aquatic organisms were use?
Lines 76 and 77 “The nutrient solution was taken from the plant section of the system itself, which had been in operation for 4 months at that point.” The nutrient solution was taken from the aquatic system?
A nutrient profile of water taken from the aquatic system needs to be shown.
Finally, consider the following comment. The presented study of the effect of pathogens on lettuce plants can be carried out in plants grown in soil or hydroponics. What is the relevance to consider an aquaponic system? This is not justified in the manuscript.
See comments on the manuscript

Minor editing of English language required
Author Response
Dear Reviewer,
We wanted to take a moment to express our sincere gratitude for the time and effort you invested in reviewing our manuscript titled "Fungal Diseases Of Cultivated Plants In Aquaponic Systems – Prevention By VIR-NIR Spectroscopy Monitoring." Your insightful comments and constructive criticisms were invaluable in enhancing the quality of our work.
Answers to the questions and comments posed:
A scheme of the system is necessary to understand. From the information it is not possible to evaluate if the system was an open or close one.
The system was closed, and water from the aquaponic system (post-biological filtration in the plant section) was added to the cultivation vessels.
It is described that plants with three leaves were placed in tanks. In other section it is said that plants in pots were taken out to the system for spectrophotometry measurements. It is confusing. How was the system. A figure may help.
A figure explaining the experiment has been added to the manuscript.
The aquatic organisms are not described. How much biomass of aquatic organisms were use?
In this experiment, common carp were cultivated in an experimental, small-scale, laboratory aquaponic system. The biomass of the cultivated fish varied between 37.7 and 40.5 kg during the trial.
Lines 76 and 77 “The nutrient solution was taken from the plant section of the system itself, which had been in operation for 4 months at that point.” The nutrient solution was taken from the aquatic system?
Yes, the nutrient solution was sampled from a long-term-operated aquaponic system.
A nutrient profile of water taken from the aquatic system needs to be shown.
The nutrient profile of the water taken from the aquaponic system was not the focus of the current study, as the same water was used for all the plants examined, both infected and control.
Finally, consider the following comment. The presented study of the effect of pathogens on lettuce plants can be carried out in plants grown in soil or hydroponics. What is the relevance to consider an aquaponic system? This is not justified in the manuscript.
The choice to study the effect of pathogens on lettuce plants within an aquaponic system is justified by the unique ecological and nutrient interactions present in this system, which can influence plant-pathogen dynamics differently compared to soil or hydroponics. By examining these interactions, we aim to contribute insights that could be relevant to both aquaponic practices and broader agricultural contexts.
See comments on the manuscript
All your comments appeared in manuscript were took in to account.
Having thoroughly addressed the questions and comments you posed, we sincerely hope that the revised article now meets your expectations. We endeavored to ensure that all your concerns were dealt with comprehensively, and the changes were incorporated in light of your valuable suggestions.
Once again, thank you for your meticulous review and for providing such constructive feedback. We look forward to any further comments or recommendations you might have, and we remain hopeful that the revised submission will meet your approval.
Sincerely,
Sirakov et al.,

Reviewer 4 Report
The aim of the study for the ms microorganisms-2508099 was to apply the visible-near-infrared spectroscopy and vegetation indices approach to test an aquaponically cultivated lettuce infected with different fungi pathogens.The authors investigated an interesting topic, but the authors should revise their ms based on my comments below.
L13: Add L. after the Latin name of Lactuca sativa
L16 revise the following text: In this study WERE included reference fungal strains ..
In the abstract, please add some key data and values for the results of your study.
Please define all abbreviations in the first mention where possible.
L47-57 merge these two paragraphs together.
L51: Three citations are enough for this statement. Also, follow the format of the citation within the text.
L54 what are these different methods of treating pathogens? Please mention them.
L58-63: please add suitable citation for the following text; Spectral features of plants in the visible region from 400 58 to 700 nm are connected mainly with content of pigments. The structure of leaf and their water content define the spectra in the near infrared region from 700 to 1100 nm. Spectral vegetation indices are developed based on plants reflectance at two or more wavelengths designed to highlight a particular property of vegetation. They are used to analyze and to detect changes in plant physiology and chemistry.
L65: Add L. after the Latin name of Lactuca sativa
L76 remove the space “). The”
L80 what was the nutrient solution? Mention it please.
L69-82 what was the name of the Experimental design? I do not see it!
L84 revise> In this study were included reference fungal strains
L87 and L89 potato not Potato
Please add the references where possible in the material and methods section, particularly the described methods.
L123 < 0.05. This is wrong, it should be ≤ 0.05.
L124 The IBM SPSS Statistics 26.0 package, add the company name, city of the owner company, and country after this text directly.
L126 avoid using such text at the beginning of each paragraph “The average lettuce reflectance spectra are presented in Figure 1” Instead, write directly about what you would like to write about, and at the end. middle or beginning of the paragraph add the citation of the Figures or Tables.
The discussion section should be revised and the authors should make it deeper than the current version using the different mechanisms of their treatments.
some minor revisions are needed
Author Response
Dear Reviewer,
We wanted to take a moment to express our sincere gratitude for the time and effort you invested in reviewing our manuscript titled "Fungal Diseases Of Cultivated Plants In Aquaponic Systems – Prevention By VIR-NIR Spectroscopy Monitoring." Your insightful comments and constructive criticisms were invaluable in enhancing the quality of our work.
L13: Add L. after the Latin name of Lactuca sativa
Done
L16 revise the following text: In this study WERE included reference fungal strains ..
Done
In the abstract, please add some key data and values for the results of your study.
Done
Please define all abbreviations in the first mention where possible.
The abbreviation and the name of all used vegetative indices were presented in table 1.
L47-57 merge these two paragraphs together.
We have significantly changed the content of both paragraphs in accordance with the notes from the other reviewers.
L51: Three citations are enough for this statement. Also, follow the format of the citation within the text.
To our understanding, there is no restriction on the number of studies that can be cited in papers published by MDPI. All the cited studies are relevant to the problem discussed, which is why we believe it would be valuable to retain them.
L54 what are these different methods of treating pathogens? Please mention them.
Text was added in accordance with the reviewer's notes.
L58-63: please add suitable citation for the following text; Spectral features of plants in the visible region from 400 58 to 700 nm are connected mainly with content of pigments. The structure of leaf and their water content define the spectra in the near infrared region from 700 to 1100 nm. Spectral vegetation indices are developed based on plants reflectance at two or more wavelengths designed to highlight a particular property of vegetation. They are used to analyze and to detect changes in plant physiology and chemistry.
Citation are added.
Daniel A. Sims, John A. Gamon. Relationships between leaf pigment content and spectral reflectance across a wide range of species, leaf structures and developmental stages. Remote Sensing of Environment 81, 2002, 337– 354.
Wan, L.; Li, H.; Li, C.; Wang, A.; Yang, Y.;Wang, P. Hyperspectral Sensing of Plant Diseases: Principle and Methods. Agronomy 2022, 12, 1451. https://doi.org/10.3390/agronomy12061451
L65: Add L. after the Latin name of Lactuca sativa
Done
L76 remove the space “). The”
Done
L80 what was the nutrient solution? Mention it please.
Done
L69-82 what was the name of the Experimental design? I do not see it!
A figure explaining the experiment has been added to the manuscript.
L84 revise> In this study were included reference fungal strains
The revision was made based on the reviewer's notes.
L87 and L89 potato not Potato
Done
Please add the references where possible in the material and methods section, particularly the described methods.
The aquaponics system is a new technology; therefore, methods for its investigation are relatively limited. Necessary references and methods have been cited in the text to provide appropriate context.
L123 < 0.05. This is wrong, it should be ≤ 0.05.
Done
L124 The IBM SPSS Statistics 26.0 package, add the company name, city of the owner company, and country after this text directly.
IBM SPSS Statistics 26.0 package (IBM, Chicago, IL)
L126 avoid using such text at the beginning of each paragraph “The average lettuce reflectance spectra are presented in Figure 1” Instead, write directly about what you would like to write about, and at the end. middle or beginning of the paragraph add the citation of the Figures or Tables.
Done
The discussion section should be revised and the authors should make it deeper than the current version using the different mechanisms of their treatments.
Done
Having thoroughly addressed the questions and comments you posed, we sincerely hope that the revised article now meets your expectations. We endeavored to ensure that all your concerns were dealt with comprehensively, and the changes were incorporated in light of your valuable suggestions.
Once again, thank you for your meticulous review and for providing such constructive feedback. We look forward to any further comments or recommendations you might have, and I remain hopeful that the revised submission will meet your approval.
Sincerely,
Sirakov et al.,

Round 2
Reviewer 2 Report
I think the revised version reads well and will make an impact on the scientific community
Much improved
Reviewer 3 Report
The quality of the manuscript has improved greatly.